# Comparing the performance of a large language model and naive human interviewers in interviewing children about a witnessed mock-event

Yongjie Sun[1,2☯]*, Haohai Pang[2,3☯], Liisa Järvilehto[4,5], Ophelia Zhang[6], David Shapiro, Julia Korkman[4,7], Shumpei Haginoya[8], Pekka Santtila[2,9]

1 The School of Psychology and Cognitive Science, East China Normal University, Shanghai, China, 2 New York University Shanghai, Shanghai, China, 3 Center for Data Science, New York University, New York, New York, United States of America, 4 Åbo Akademi University, Turku, Finland, 5 Forensic Psychology Center for Children and Adolescents, Helsinki University Hospital, Helsinki, Finland, 6 Maastricht University, Maastricht, The Netherlands, 7 European Institute for Crime Prevention and Control, affiliated with the United Nations (HEUNI), Helsinki, Finland, 8 Meiji Gakuin University, Tokyo, Japan, 9 Shanghai Frontiers Science Center of Artificial Intelligence and Deep Learning, New York University Shanghai, Shanghai, China

☯ These authors contributed equally to this work.
* ys6261@nyu.edu

## Abstract

### Purpose

The present study compared the performance of a Large Language Model (LLM; ChatGPT) and human interviewers in interviewing children about a mock-event they witnessed.

### Methods

Children aged 6-8 ($N = 78$) were randomly assigned to the LLM ($n = 40$) or the human interviewer condition ($n = 38$). In the experiment, the children were asked to watch a video filmed by the researchers that depicted behavior including elements that could be misinterpreted as abusive in other contexts, and then answer questions posed by either an LLM (presented by a human researcher) or a human interviewer.

### Results

Irrespective of condition, recommended (vs. not recommended) questions elicited more correct information. The LLM posed fewer questions overall, but no difference in the proportion of the questions recommended by the literature. There were no differences between the LLM and human interviewers in unique correct information elicited but questions posed by LLM (vs. humans) elicited more unique correct information per question. LLM (vs. humans) also elicited less false information overall, but there was no difference in false information elicited per question.

**Data availability statement:** The data are held or will be held in a public repository at https://osf.io/5whc7/?view_only=9377fd48c46a-412e95a23d08b1e846c3.

**Funding:** This work was supported by the Sundell Foundation and the Shanghai Frontiers Science Center of Artificial Intelligence and Deep Learning at NYU Shanghai.

**Competing interests:** The authors have declared that no competing interests exist.

## Conclusions

The findings show that the LLM was competent in formulating questions that adhere to best practice guidelines while human interviewers asked more questions following up on the child responses in trying to find out what the children had witnessed. The results indicate LLMs could possibly be used to support child investigative interviewers. However, substantial further investigation is warranted to ascertain the utility of LLMs in more realistic investigative interview settings.

## Introduction

Children are interviewed in diverse legal contexts, such as investigations of child abuse, child trafficking, asylum procedures and custody disputes. In many of these, the child's account is crucial to the investigations given that their statement may be the only evidence [1–3]. When interviewing children in these contexts, it is important to have an information-gathering/hypothesis-testing approach rather than attempt to confirm a particular assumption about what may have happened [4,5]. Unfortunately, research shows that the quality of these interviews is often substandard [6]. Despite efforts to improve interview quality through training programs, challenges remain, wherefore attempts to test the possible use of AI to improve the quality of interviews are warranted. Recent advances in Instruction Tuning and Few-Shot Learning technologies have enabled LLMs to show consistent and reliable performance across various professional domains, including medical diagnosis and legal investigation. Given our anticipation that LLMs would exhibit superior adherence to interview protocols, we hypothesized that LLMs would show enhanced performance compared to human interviewers in simulated child eyewitness interviews. Here, we tested the ability of a large language model (LLM, specifically ChatGPT; OpenAI, 2023) to ask questions in real time during an interview of children about a witnessed mock-event. The performance of the LLM was compared to that of naive human interviewers.

Ample evidence suggests that only increasing knowledge of how children should be interviewed (i.e., using open rather than closed questions) is not sufficient to change interviewers' over-reliance on closed questions [7]. In fact, one of the main reasons behind the development of structured investigative interviewing protocols was the need to mitigate the effects of suggestibility on the quality of the narratives elicited from children during these interviews [8]. Lamb et al. [9] have shown that using structured protocols in combination with extensive feedback can lead to impressive improvements in interview quality. Brubacher et al., [10] also found that intensive learning maintained the learning skills, and the effect can last over one and a half years during the interrogation course [11,12]. Interviewing training programs enhance the interviewing performance of interviewers with varying experience levels, specifically improving use of supportive utterances, adherence to a child-appropriate interviewing style and the quality of questions being asked [13–16]. However, the positive effects can diminish rapidly once feedback is discontinued [9] and providing such feedback continuously may be logistically and cost-wise challenging. Furthermore, the lack of outcome feedback on the actual veracity of child responses in real interviews may hamper interviewers' awareness of potential false details they have elicited, impeding their motivation to modify their interviewing techniques [17].

Research consistently supports the use of open-ended questions (e.g., "Tell me what happened.") as the most effective way to obtain reliable information from children [18–21]. The National Institute of Child Health and Human Development (NICHD) Investigative Interview

Protocol represents one of the most extensively validated semi-structured approaches for conducting forensic interviews with children [8,22–24]. The protocol progresses through distinct phases: introduction and ground rules, rapport building, practice narrative, substantive phase, and closure [21,25]. Central to its effectiveness is the emphasis on open-ended prompts (See Table 1 for details) that elicit free-recall narratives from children, as these yield more accurate and detailed responses compared to focused or closed questions [22,26]. Even experienced interviewers do not, however, use these types of questions in their interviews as much as is recommended [27–29]. They may also begin with recommended questions but move on to more directive questions earlier that would be advisable [30]. Research offers some insight into why this is. For example, professionals often deviate from this approach due to the need for specific investigative details, unfamiliarity with open-ended techniques, difficulty distinguishing between question types or over-reliance on common sense beliefs [31,32]. Additionally, even leading questions are sometimes seen as necessary by experienced interviewers, especially in challenging interviews, and sometimes due to outside encouragement from other professionals involved in the case [33]. There can also be organizational barriers, such as lack of resources, or even a lack of confidence in best evidence provided by scientific research, that can make adoption of best practices less likely [34].

To address these limitations, researchers have explored the efficacy of simulated avatar interview training programs and other computer-based learning techniques [5,13,28,35]. A mega-analysis conducted by Pompedda et al. [36] on one variant of such training revealed encouraging results in improving interview quality. However, such training programs may not be universally effective due to individual differences in response to the interventions. For example, Krause et al. [37] found that only 32-43% of participants in the two intervention groups showed a reliable change in their interviewing performance even though the group level statistics evidenced reliable improvements. Also, as with all training programs, learned skills in interviewing do not always transfer well into actual interviews, and there are individual differences in the transfer rates [38]. David and Lorraine [38] showed that transfer of police training to field practice is influenced, both directly and indirectly, by multiple factors including trainee motivation, perceived training relevance and quality, and task preparedness.

**Table 1. Detailed Information on Question Types.**

| Question Category | Definition | Examples | Recommended Use | Effect on Child's Response |
|---|---|---|---|---|
| Invitations/Open-ended prompts | Questions that allow free recall and give children control over disclosure | • "Tell me everything that happened"<br>• "Tell me more "<br>• "What happened next?" | Primary questioning strategy; should be used as much as possible | • Longest responses<br>• Most accurate information<br>• Most detailed accounts<br>• Child has control over narrative |
| Cued invitations | Open questions that follow up on details already mentioned by the child | • "You mentioned [detail]. Tell me more about that"<br>• "You said [event]. Tell me everything about that" | Should be used frequently to explore details mentioned by child | • Helps maintain focus while remaining open<br>• Elicits elaboration on specific topics<br>• Based on child's own words |
| Directive questions | Focused questions about previously mentioned information (who, what, when, where) | • "When did that happen?"<br>• "Where were you?"<br>• "Who was there?" | Use only when open-ended questions cannot elicit needed information | • More focused but still allow some elaboration<br>• Generally accurate |
| Option-posing/Forced choice | Questions providing specific options or requiring yes/no answers | • "Was it day or night?"<br>• "Did he hit you?"<br>• "Were your clothes on or off?" | Should be limited; use only when necessary details cannot be obtained through other means | • Very brief responses<br>• Risk of guessing - Lower accuracy<br>• May lead to acquiescence |
| Suggestive questions | Questions containing information not mentioned by child or implying desired response | • "He hit you, didn't he?"<br>• "That must have hurt"<br>• "Did he do other bad things?" | Should be avoided | • May lead to child providing false information during the interview and later on<br>• Can compromise testimony credibility<br>• Highest risk of inaccurate responses |

In the context of open-ended questioning, Melinder and Gilstrap [39] established a positive correlation between officers' self-regulatory capacity and their ability to employ open-ended questions effectively. Furthermore, research has explored the relationship between individual characteristics and overall investigative interviewing outcomes with children. Al Ali et al. [40] revealed significant associations between emotional intelligence and investigator performance. Personality factors, specifically openness, conscientiousness, and neuroticism, have also been identified as significant predictors of child interviewing effectiveness [15,41].

In fact, evidence for transfer from avatar interviews to actual interviews is yet limited [42]. Also, implementing any training program is costly and logistically challenging. Training investigators in investigative interviewing of children is also challenging as conducting such investigations is demanding and stressful for the police officers involved in child abuse investigations [43–45]. Job related stress is likely to lead to turnover rates that produce a constant need to train new professionals in forensic interviewing. Finally, delivering the required training programs requires continued commitment and resource allocation from the leadership of the relevant organizations.

Another challenge, not given adequate attention in the investigative interviewing context previously, is that training cannot eliminate variations in interview quality caused by situational factors: investigators can become tired, their level of motivation can fluctuate, they can become distracted or unable to keep in mind the relevant case related background information and interview objectives. Their personality [46] and emotions [47] can also have an effect on the quality of the interview. Research is scarce on how these factors affect the quality of the questions asked by the interviewer, but quality hampering mechanisms have been proposed. The existing information points to investigative interviewing being a demanding endeavor with high cognitive load associated with it [48,49]. Investigative interviewing involves several cognitive processes, both intrinsic and extraneous in nature, putting demands on the attention of interviewers. This encompasses not only the task of retaining pertinent information but also the strategic orchestration of the interview itself. Interviewers need to apply their accumulated knowledge in a proactive manner, which involves listening to witnesses, retaining the substance of their statements, attending to the unique needs and communication styles of each witness, phrasing questions which aid in assessing the veracity of the narratives, selecting appropriate inquiry techniques, delineating subjects to pursue, and soliciting clarification from witnesses [50]. Moreover, in investigations involving interviews with multiple witnesses, interviewers need to monitor the provenance of the information elicited to enable effective differentiation between the various accounts.

The Self-Administered Interview (SAI) is a widely used self-report tool for witness interviewing. While it allows witnesses to provide information without third-party interference and can elicit more accurate details [51], the self-reported methods also have limitations, especially with children. For instance, using this tool requires individuals to have well-developed linguistic and cognitive abilities. Therefore, younger children, whose language and cognitive abilities are still developing, might find it challenging to use such self-reporting methods effectively [52,53]. Additionally, Del Castillo and Wright [54] pointed out that children tend to disclose to people with whom they have close relationships, but self-report questionnaires cannot meet this relational requirement. Therefore, it is important to have interviewers present appropriate questions to children.

It was also not completely clear whether experience in interviewing children affects questioning quality. Adults often use closed questions in non-investigative interactions with children [55] meaning that investigative interviewing may require a departure from normal modes of interaction. Also, the usual lack of knowing the ground truth in alleged CSA cases, and the circular use of legal outcomes as proof of interview quality might maintain less than

optimal practices [56]. In fact, research results are unclear with field studies showing no necessary association between experience and open question use [e.g., 30], with some exceptions [e.g., 15]. Moreover, there is even some evidence of a negative association between experience and open question use [e.g., 28] in experimental studies.

Studies have consistently showed that children's initial disclosures of abuse often occur to non-professional individuals rather than trained forensic interviewers. In examining first disclosures, Magnusson [57] found that among 57 abused children aged 3–7 years, over 10% chose to initially disclose to their kindergarten teachers, significantly higher than first disclosures to police investigators and psychologists (2.6%). Similar patterns of initial disclosure were observed by Münzer et al. [58], with first disclosure rates of 10% to social workers and 5% to teachers. In terms of overall disclosure patterns, larger-scale studies have reinforced this trend. Lev-Wiesel et al. [59] reported that over 55% of 281 Israeli children disclosed to non-professionals including teachers, counselors, and social workers. Similarly, Postmus et al. [60], in a study of 708 abuse disclosure cases, found that 35% disclosed to teachers, 33% to community authority figures, 30% to school staff, and 30% to religious leaders. Among formal recipients, 33% disclosed to healthcare providers and 19% to counselors. This pattern of disclosure, particularly the prevalence of initial disclosures to non-professionals, suggests potential applications for LLM-based interviewing systems. Furthermore, it provides theoretical justification for including naive participants as a comparative baseline in examining LLM interviewer performance, as they represent a significant portion of real-world disclosure recipients.

Considering (a) the limitations of current training approaches in terms of costs, logistics and organizational challenges and (b) the inherent cognitive demands of the task, an alternative strategy could be providing direct support to interviewers in real time during ongoing interviews in the form of suggesting a question that the interviewer could ask next. By offering such suggested questions in real-time, the identified challenges may be solved. Recent developments in artificial intelligence and especially in large language models (LLMs) may offer an avenue to achieve this type of real time assistance.

## Large language models (LLMs)

Large language models (LLMs) refer to advanced natural language processing systems trained on vast amounts of text data, capable of determining the next segment of generated language based on the semantics of the input. LLMs' ability to analyze and generate language data at an unprecedented scale have the potential to transform research in many fields of psychology. By simulating human language skills without human cognition or emotions, LLMs offer a tool for simulating realistic interactions [61], including the nuances of child investigative interviewing. LLMs' ability to analyze and generate vast amounts of language data has the potential to add to the ways we currently study psychological constructs, including the nuances of child investigative interviewing. In the past years, the integration of conversational Artificial Intelligence systems has increasingly become an interest across various disciplines, including the healthcare sector [62].

Currently, OpenAI's ChatGPT is one of the most developed large language model systems. In the training and application of large language models, one can provide explicit instructions for its subsequent language generation through prompt-engineering, allowing it to produce varying information based on different situations and requirements. Research in other contexts suggest promise in using LLMs in interviewing tasks. Røed et al. [63] and Hassan et al. [64] fine-tuned the GPT-3 model to simulate a child interviewee by using interview data with the model being able to respond flexibly and realistically to questions posed. This use of the LLM overcame the shortcomings of previous Avatar studies where the answers were fixed. Neither study explored the possibility of using LLMs to formulate questions to the child, that

is, acting the role of the interviewer rather than the interviewee. In another context, Pollina and Barretta [65] found that computer-generated agents could elicit more information than self-report regarding adult applicants' drug, alcohol and criminal history in a security screening setting. In a review, Noriega [66] concluded that LLMs' racial and gender biases were lower than those of human interrogators. Moreover, LLM interrogators had an advantage in building relationships and obtaining information compared to human interviewers. However, to the best of our knowledge, LLMs have not been yet tested in the context of investigative interviewing of children.

Given that LLMs are unlikely to suffer from the challenges that human investigative interviewers have in keeping to guidelines (even though their training data may reflect some of these features), it is possible that LLMs will be superior to especially naive human interviewers in terms of formulating open questions.

Given that LLMs are the result of pre-training on vast amounts of data, it may prove challenging for them to extract relevant information from their training data and respond effectively. However, Instruction Tuning and Few-shot Learning techniques effectively improved this challenge. Through Instruction Tuning [67], the model learns to follow specific guidelines and principles, allowing it to focus on relevant knowledge rather than being overwhelmed by its broad training data. When combined with Few-shot Learning [68], the model can quickly adapt its vast knowledge to the specific requirements. Specifically, the example enables the model to extract interviewing patterns and behavioral protocols from demonstration cases [69], while Task Clarification establishes operational parameters and performance criteria for the interview process [70,71]. This combination enables the model to generate appropriate questions that align with professional standards, effectively bridging the gap between its broad training and specific task requirements. Consequently, many scholars have introduced the concept of prompt-engineering. Specifically, this involves providing pre-trained LLMs with instructions, or prompts, that guide the model to focus the subsequent conversation within a specific domain [72,73]. By doing so, the performance of LLMs can be enhanced. Chen et al. [73] and Park et al., [74] found that the naturalness and specificity of the sentences generated improved in the prompt-engineered models compared to unmodified models. Wang et al., [75] provided standardized diagnostic guidelines as a system prompt to LLMs, then offered detailed information about a case in each conversation, asking LLMs to make judgments based on the information provided and to repeat the process multiple times. The results found that the advanced model with a strong (vs. moderated and limited) level of prompt performed with a reliability of over 75% in rounds of tests. Walker et al., [76] also found that the consistency of the answers provided by LLMs with the suggested answers reached 60%. The consistency between LLMs' judgments and humans was.78, and the internal consistency of multiple rounds by LLMs reached 100%, suggesting high stability of prompted LLMs.

## Aims and hypotheses

Based on the expectation that LLMs will better adhere to the interviewing guidelines [75,76] while naive human interviewers may not be able to do so [27–30], we formulated the following exploratory hypotheses:

Hypothesis1: Question Formulation: We expected the LLM (vs. naive human interviewers) to formulate (a) more recommended questions (e.g., directive questions, "Tell me what happened."), both in absolute numbers and proportionally, and (b) less non-recommended questions (e.g., close-ended questions, "Did he touch the girl?").

Hypothesis 2: Information Elicitation: We expected the LLM (vs. naive human interviewers) to elicit (a) more correct information and (b) less incorrect information from the children.

Based on previous research indicating that recommended questions can trigger children's free recall, whereas not-recommended questions may lead to false memories due to their suggestive nature [22,26], the following hypothesis was also tested:

Hypothesis 3: Irrespective of the interviewer type, we expected recommended (e.g., tell me what happened.) questions to elicit (a) more new correct information and (b) less incorrect information from the children compared to not recommended questions (e.g., did he touch the girl?)

## Method

This experiment was approved by the New York University (Shanghai)'s Institutional Review Board (2022-035-NYUSH-New Bund). Fully-informed consent was obtained from at least one parent or guardian through a web-based consent form and oral assent was obtained from the child. The researcher explained the whole procedure to the child and reminded the child that they could withdraw at any time without any negative consequences. Additionally, the parents were informed that they were free to monitor the interview process and end it at any time. The adult human interviewers also provided fully-informed consent via a web-based consent form.

### Participants

Participant recruitment began on March 6, 2023 and ended on October 2, 2023. A total of 78 children ($M_{age}$ = 6.95, $SD$ = 0.82) aged from 6 to 8 were recruited. The median of age was 7. Participants scheduled sessions independently through separate registration systems. Children were assigned to either human-conducted interviews (when both child and adult participants booked the same slot, $n$ = 38, $M_{age}$ = 6.97, $SD$ = 0.82, including 21 (52.5%) male) or LLM-conducted interviews (when only a child booked the slot, $n$ = 40, $M_{age}$ = 6.93, $SD$ = 0.83, including 22 (57.9%) male), ensuring random assignment to experimental conditions. There was no difference in the ages of the two groups, $t(76)$ = .26, $p$ = .795, nor in the genders, $\chi^2 (1)$ = .23, $p$ = .401.Thirty-eight university students, fluent in Mandarin, were recruited to serve as interviewers in the human interview group.

Hedges [77] suggests that an effect size ≤ 0.2 can be considered small, 0.5 medium and 0.8 a large effect, and this criterion could also be employed in investigative interviews with children [23]. Based on this, a power-analysis with G*POWER showed that, assuming an effect size of.50 with α = .05, the sample size would result in a power of.706.

### Materials

This study presented the events in video format for the children which means that their task resembled that of a witness rather than a victim. Two authors created six videos with different variations in child gender and the interaction elements for the children to memorize. Specifically, the interaction with the children was conducted by one of the authors (H. P.), while another research assistant handled the camera for filming. The video recording took place in a quiet environment. The setup in the room was consistent across videos. Several videos were created in order to vary the to-be-remembered events to increase the generalizability of the findings and to thus exclude any between-session learning by the LLM. The videos lasted 30 seconds to 1.5 minutes. Each video featured an adult male and a child interacting in a room. During the experiment, one video was randomly presented for each child to watch and review.

Elements in the videos included behaviors that could be present in abusive contexts such as taking photos, physical contact (to assist that child to take a photograph and show them the photograph taken), and also neutral behaviors like conversations. Please, see S1 Table for details of the video contents.

## Procedure

The experiment was conducted online via the Tencent Meeting platform. When children registered for the experiment, they and their parents were briefed and consent was elicited from the parent while assent was elicited from the child. The parent could choose to monitor the whole experimental procedure with a strict instruction not to assist the child in responding to the questions. Next, the children were randomly assigned to either the LLM or human interviewer group and they were not informed whether the questions asked were formulated by LLM or the human interviewer. Prior to the interview, they viewed one of the stimulus videos. In the LLM group, questions were generated in real-time by inputting the child's response into a prompt-engineered ChatGPT-3.5 model by an assistant and after the LLM generated the next question, the assistant read the questions to the child; in the human interview group, the naive human interviewers posed the questions on the spot. Both the LLMs and human participants could decide to stop when they thought they had elicited sufficient information. The human interviewers signaled the researchers to terminate the session. For LLMs, explicit termination responses provided by the LLMs, such as "I think I have gathered enough information" were considered indicators of conclusion. Questions that serve as markers of termination were not posed to the children. Instead, the children were asked, "Do you think these questions were asked by me or generated by a computer?".

Regarding the parameter settings of the LLM applied in this study, since the research directly utilized OpenAI's open web interface (https://chat.openai.com/) rather than the API, the model's parameter settings are the default settings provided by OpenAI. OpenAI has not provided detailed information about the parameter settings for ChatGPT-3.5.

In the human interview group, only after the child had watched the video, did the human interviewer join the meeting to start the interview, ensuring that the interviewer was unaware of the video's content. Both human interviewers and LLMs received the same set of guidelines (prompt) (see Table 2). The entire interview process was recorded.

**Table 2. Guidelines (Prompt) Given to Both LLMs and Human Interviewers.**

*I want you to play the role of an interviewer. I am going to play the role of an 8-year-old child (There will be another child) [Role Playing, as suggested by instruction tuning [78]]. Your task is to ask me (the child) questions, to find out what happened in the video I (the child) just watched [Task Definition [70,71], as suggested by prompt engineering].*

*Guidance: Abused children are often reluctant to talk about what they have experienced when they are interviewed by police officers due to shame and embarrassment. Also, if the interviewer uses leading questions, children are likely to give answers to such questions that contain inaccurate information [Example Passes [69], as suggested by few-shot learning]. I (The child) have viewed a video in which a child played a game with an adult. Your task as the interviewer is to figure out what happened in the video [Task Definition [70,71], as suggested by prompt engineering]. Although there is no specific script for the interview, no questions should go beyond the purpose of finding out what happened in the game. Questioning about my (the child's) personal information is NOT allowed [Constraint Setting [79,80], and for data security].*

*The game might have involved conversation and photographs taken. Your task is also to get a description of the adult who participated in the game. Please try to always use open questions. I want you to do the interview by asking one question at each time and wait for my (the children's) response [Providing specific instructions]. Then, you may keep asking until you feel comfortable about what happened [Stop Condition].*

Note. The underlined section with a horizontal line indicates the differences between what LLM and human participants received. The content within the brackets on the horizontal line is the instruction content for the human interview group, while the content outside the brackets is for the LLM interview group. Apart from the section with the horizontal line, all other content was identical.

Text enclosed in square brackets explains the rationale for including the preceding sentence in the prompt. These bracketed explanations were not presented to either LLMs or human interviewers.

## Coding

**Question category coding.** The coding system was based on the meta-review on the NICHD Protocol by Benia et al. [23] and Lyon [81] which compared open-ended, option-posing and close-ended questions. Furthermore, our coding scheme for detail categories was based on the classifications outlined in Sternberg et al. [82] and Lamb et al. [83], and we adopted analogous categories: The "unrelated" results from "Non-substantive", "open-ended" from "Invitations" and "Directive", and "close-ended" from "Suggestive". Additionally, the categorization was in basic agreement with the scheme presented by Haginoya et al. [84], Pompedda [85] and Waterhouse et al. [86]. Based on the above references, this study classified questions into six categories: Unrelated, Too Long, Repeated, Directive, Closed, Open. Directive and Open were categorized as Recommended, and the other four categories were classified as Not Recommended. The proportion of Recommended questions of the total number of questions in each interview was calculated, named Recommended Question Rate. See Table 3 for definitions of all question categories.

**Answer coding.** During an investigative interview, the information provided by the interviewee can fall into three categories: (1) Correct information, where the provided details align with the facts of the case; (2) Incorrect information, where the general context of the information matches reality, but the specifics do not; and (3) Confabulated information, where the scope of the details does not align with reality or includes irrelevant categories. For example, consider a scene containing a single white cup on the table. Recall responses were categorized as follows: statements such as "there was a white cup on the table" were classified as correct information; statements like "there was a black cup on the table," where the object was correct but details were inaccurate (black versus the actual white), were classified as incorrect information; and statements such as "there was a white cup and a phone on the table" contained confabulated information, where mentioned objects (the phone) were not present in the original video scene. Correct information is genuinely valuable, providing targets for subsequent investigations. In contrast, incorrect and fabricated details can slow down, or even misdirect future investigations, leading to increased investigative costs and reduced efficiency.

Key elements in the videos were first determined and listed before the answer coding. All key elements in a video were divided into environmental elements and character elements.

Environmental elements were defined as follows: They were static elements following the "category (detail)" format, for example, chair (black). Character elements were divided into static and dynamic. Among them, static elements also followed predetermined judgment criteria. Static information about the adults included gender (male), attire (light-colored

**Table 3. Question Categories.**

| Category | Sub-Category | Content | Example |
|---|---|---|---|
| Not Recommended | Unrelated | The question is unrelated to the scenario. | "How's your relationship with your mom?" |
| | Too Long | More than two questions at a time asked. | "How did the child and the adult in the video choose the phone? What did they say?" |
| | Repeated | The question is the same as or similar to the previous question. | "What would he do?" "What was he going to do?" |
| | Closed | A yes-no question or a question providing two or more options. | "Did he wear a hat?" "Is the hat black or white?" |
| Recommended | Directive | The question includes some indicative words, such as "what", "when", or "how". | "What did they photograph?" |
| | Open | Questions that cannot be answered using a single word. | "Tell me about his hat" |

t-shirts and shorts, hats, and eyeglasses) and the same about the child present in the video. The appearance of both the adult and the child were not coded because this information was ambiguous rather than objective. Dynamic elements of characters included the [subject], [action], [tool], [object of action], and [content of action]. Each content detail was calculated as a correct piece of information. The detailed coding plan for the two kinds of elements was shown in S2 Table.

The total number of correct and incorrect pieces of information mentioned by children in each interview was tallied. According to the coding principles of this study, information repeatedly mentioned by children during these repetitions was also recorded under correct information. To separate between new and already mentioned information, a "Unique information" variable was introduced in the coding, which represented the total number of unique correct pieces of information mentioned by the child without repetitive counting of the same detail. An "information rate" variable was added, representing the ratio of information to the total number of questions, indicating the amount of correct information that was obtained per question. This variable can be understood as questioning efficiency. Additionally, a "Proportion of unique information" variable was calculated by the number of unique details reported by children divided by the total number of unique details in the video. The variables and their definitions are listed in detail in Table 4.

## Coding of questions and answers

After the experiment, all videos were transcribed using the open-source software Buzz [87]. Once transcription was complete, initial data cleaning was done, mainly to match questions and answers and to remove filler words. The questions and answers were coded by two independent coders who were initially trained by the first author. They were also blind to the experimental hypotheses and conditions and had no prior involvement with the study. Reliability

**Table 4. Variables Coded from Child Answers.**

| Variable | Calculate |
|---|---|
| Question Source | The children were asked whether the questions were proposed by humans or AI after the interview in the LLM group.<br>0 = the child thought the questions were posed by human<br>1 = the child thought the questions were posed by AI |
| Interviewer Type | 0 = LLM<br>1 = human interviewer |
| Number of Questions | The total number of questions in an interview |
| Correct Information | How many correct details were mentioned in each interview. |
| Incorrect Information | How many incorrect details were mentioned in each interview. |
| Confabulated Information | How many confabulated details were mentioned in each interview. |
| False Information | = Incorrect Information + Confabulated Information |
| False Information Rate | = (Incorrect Information + Confabulated Information)/ Number of Questions<br>How much incorrect and confabulate information was obtained in each Q-A round |
| Unique Information | How many unique correct details were mentioned in each interview (Compared with Correct Information, Unique Information only counted each new detail once) |
| Information Rate | = Unique Information/ Number of Questions<br>How much unique correct information was obtained in each Q-A round. |
| Proportion of Unique Information | = Unique Information/ Total number of details in each video<br>Of all the details in the video, how much did the children report. |

on initial training for 150 question-answer pairs was.76, for correct information it was.94, for false information it was.84, $p < .001$. These values suggested excellent inter-rater reliability. The order in which the interviews were coded was randomized based on the child's ID number to minimize bias during coding and each of the coders completed half of the data.

## Statistical analyses

Data analyses were conducted using SPSS 27.0. Given the strong correlations among our primary dependent variables (See Table 5 for details), a Multivariate Analysis of Variance (MANOVA) was conducted with Bonferroni correction. For dichotomous dependent variables (question authenticity), a non-parametric Mann-Whitney U test was employed.

## Results

### Descriptive analyses

First, descriptive analyses were conducted. There were 1343 question-answer pairs in total. Of these, 471 were posed by the LLMs and 872 were posed by humans. See Table 5 for other descriptive statistics.

S3 Table shows one example interview transcript and the coded question categories and details from each of the two groups.

First, we checked for differences between the two groups. The LLM ($M = 10.78$, $SD = 4.84$) asked fewer questions than the human interviewers ($M = 22.95$, $SD = 16.56$), $F (1, 76) = 19.48$, $p < .001$, $\eta^2 = .204$. This is a medium effect. The interview with the fewest questions included only 3 questions. In this interview, the child's single response provided a significant amount of information, and the interviewer believed that the information was sufficient. Therefore, this interview was not considered an outlier. The interview with the most questions had 60 questions. Both of these interviews were from the human interviewer group.

Table 6 shows the numbers and proportions of the different question types in the LLM and human interviewer conditions. There were significant differences between LLM and the human interviewer in the types of questions they asked, $X^2(5) = 80.91$, $p < .001$. Comparison of the proportions by question type showed that LLM (vs. naive human interviewers) asked less unrelated, more too long, less repeated, more directive, and less open questions, while there was no significant difference in the number of closed-questions posed by LLM and human. See Fig 1 for details.

### Hypothesis 1: quality of questions posed by LLM and human interviewers

First, the number of question types was analyzed based on the originality. Results showed that LLM posed significantly less recommended questions ($M = 3.18$, $SD = 2.34$) compared to

**Table 5. Descriptive Analyses and Correlations.**

| | M (SD) | | | | | | | |
| --- | --- | --- | --- | --- | --- | --- | --- | --- |
| | Humans | LLMs | 1 | 2 | 3 | 4 | 5 | 6 |
| 1 Number of Questions | 22.95 (16.56) | 10.88 (4.90) | 1 | | | | | |
| 2 Number of Recommended Questions | 5.63 (5.02) | 3.18 (2.34) | .763** | 1 | | | | |
| 3 Number of Not-Recommended Questions | 17.32 (13.18) | 7.70 (3.76) | .969** | .581** | 1 | | | |
| 4 Number of Correct Detail | 23.00 (21.19) | 14.90 (9.31) | .614** | .718** | .501** | 1 | | |
| 5 Number of False Detail | 2.84 (3.57) | 1.73 (1.55) | .623** | .607** | .555** | .484** | 1 | |
| 6 Number of Unique Information | 7.50 (4.74) | 8.50 (3.57) | .344** | .494** | .246* | .757** | .266* | 1 |

**Table 6. Differences in Number and Proportions of Different Question Types between LLM and Human Interviewer Conditions.**

| | | | LLMs | Humans |
|---|---|---|---|---|
| Recommended | Directive | | 126a | 187b |
| | | | 29.0% | 21.4% |
| | Open | | 1a | 27b |
| | | | 0.2% | 3.1% |
| Not-Recommended | Unrelated | | 2a | 81b |
| | | | 0.5% | 9.3% |
| | Too long | | 27a | 22b |
| | | | 6.2% | 2.5% |
| | Repeated | | 5a | 52b |
| | | | 1.1% | 6.0% |
| | Closed | | 274a | 503a |
| | | | 63.0% | 57.7% |

Note. Rows with different subscripts differ significantly from each other at the .05 level.

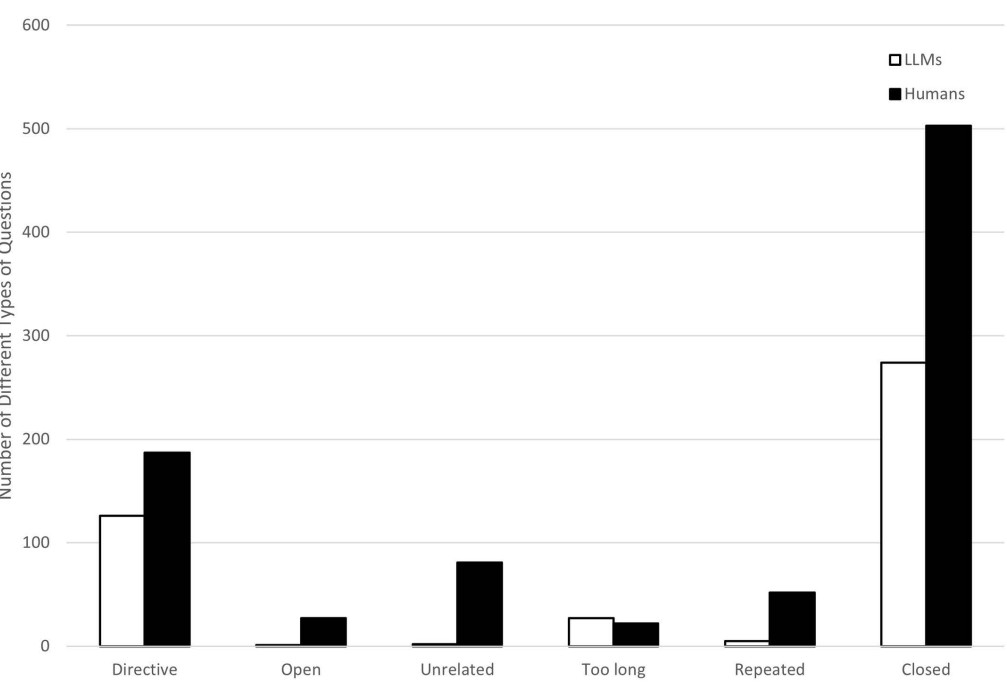

**Fig 1. Group Comparison in Question Categories.**

human interviewers ($M = 5.63$, $SD = 5.02$), $F (1, 76) = 7.80$, $p = .007$, $\eta^2 = .09$. The number of not recommended questions generated by LLM ($M = 7.70$, $SD = 3.76$) was significantly lower than by humans ($M = 17.32$, $SD = 13.19$), $F (1, 76) = 19.61$, $p < .001$, $\eta^2 = .21$. The results rejected hypothesis 1a while supporting hypothesis 1b.

Further, we examined the difference in the proportion of question types. The proportions were obtained by dividing the respective numbers by the number of questions. In the number of questions, the question about authenticity was excluded. Results showed that there was no

significant difference in the proportion of recommended questions generated by LLM ($M$ = .29, $SD$ = .18) and formulated by humans ($M$ = .27, $SD$ = .15), $F$ (1, 76) = .32, $p$ = .576. Also, there was no difference in the proportion of not recommended questions generated by LLM ($M$ = .71, $SD$ = .18) and formulated by humans ($M$ = .73, $SD$ = .15), $F$ (1, 76) = .32, $p$ = .576. The result rejected Hypotheses 1a.

The reason for the inconsistent direction of results in testing the differences in the number and proportion of the two types of questions is likely to be that the proportion is derived from dividing the number of each type of question by the total number of questions. Therefore, the difference in direction between the absolute number and relative number of the two types of questions is opposite. Overall, LLMs ask fewer questions than humans, resulting in lower absolute numbers of both recommended (e.g., Tell me what happened.) and not recommended questions (e.g., Did he touch the girl?). Therefore, the overall quality of the interviews should be assessed by comparing the proportions of recommended to not recommended questions asked by the LLM and human groups.

## Hypothesis 2: information acquired by LLM and human interviews

Next, we examined the differences between the information acquired by the LLM and the human interviewers. The total number of correct information mentioned in human interviews ($M$ = 23.00, $SD$ = 21.19) was significantly higher than in the LLM interviews ($M$ = 14.90, $SD$ = 9.31), $F$ (1, 76) = 4.86, $p$ = .030, $\eta^2$ = .06. This supports Hypothesis 2a.

In terms of false information, LLM elicited fewer pieces of such information ($M$ = 2.05, $SD$ = 1.75) compared to human interviewers ($M$ = 3.45, $SD$ = 3.90), $F$ (1, 76) = 4.25, $p$ = .043, $\eta^2$ = .05. There was no significant difference in the False Information Rate under the two conditions, $F$ (1, 76) = 1.73, $p$ = .193 This partly supported Hypothesis 2b.

## Hypothesis 3: information acquired from recommended and non-recommended questions

Next, we examined the differences in the effectiveness of the two question types in obtaining information. The amount of correct information elicited by Recommended questions ($M$ = 1.578, $SD$ = 2.346) was higher than that elicited by Not Recommended questions ($M$ = .965, $SD$ = 1.810), $F$ (1, 1303) = 24.76, $p$ < .001, $\eta^2$ = .02 supporting Hypothesis 3a. Also, the number of incorrect information elicited by Recommended questions ($M$ = .176, $SD$ = .418) was more than that of Not Recommended questions ($M$ = .121, $SD$ = .365), $F$ (1, 1303) = 9.73, $p$ = .002, $\eta^2$ = .01 not supporting Hypothesis 3b. Please see Fig 2 for detail.

## Beyond testing the priori hypotheses, we conducted a number of exploratory analyses

**Unique information acquired by LLM and Human interviews.** In testing Hypothesis 2, the correct information reported by children was calculated cumulatively, whereby repeated instances of the same correct information were counted multiple times, potentially inflating the total volume of reported information. To address this limitation, "Unique Information" variables were introduced to capture the absolute quantity of non-redundant correct information provided by children. The variables were subsequently incorporated into the MANOVA analysis. Results showed that there was no significant difference between the total number of Unique Information elicited by LLM ($M$ = 8.50, $SD$ = 3.57) and the human interviewers ($M$ = 7.50, $SD$ = 4.74), $F$ (1, 76) = 1.12, $p$ = .290. Comparing the Information Rate under the two conditions, it was found that LLM elicited significantly more unique correct information per question ($M$ = 0.90, $SD$ = 0.53) compared to human interviewers

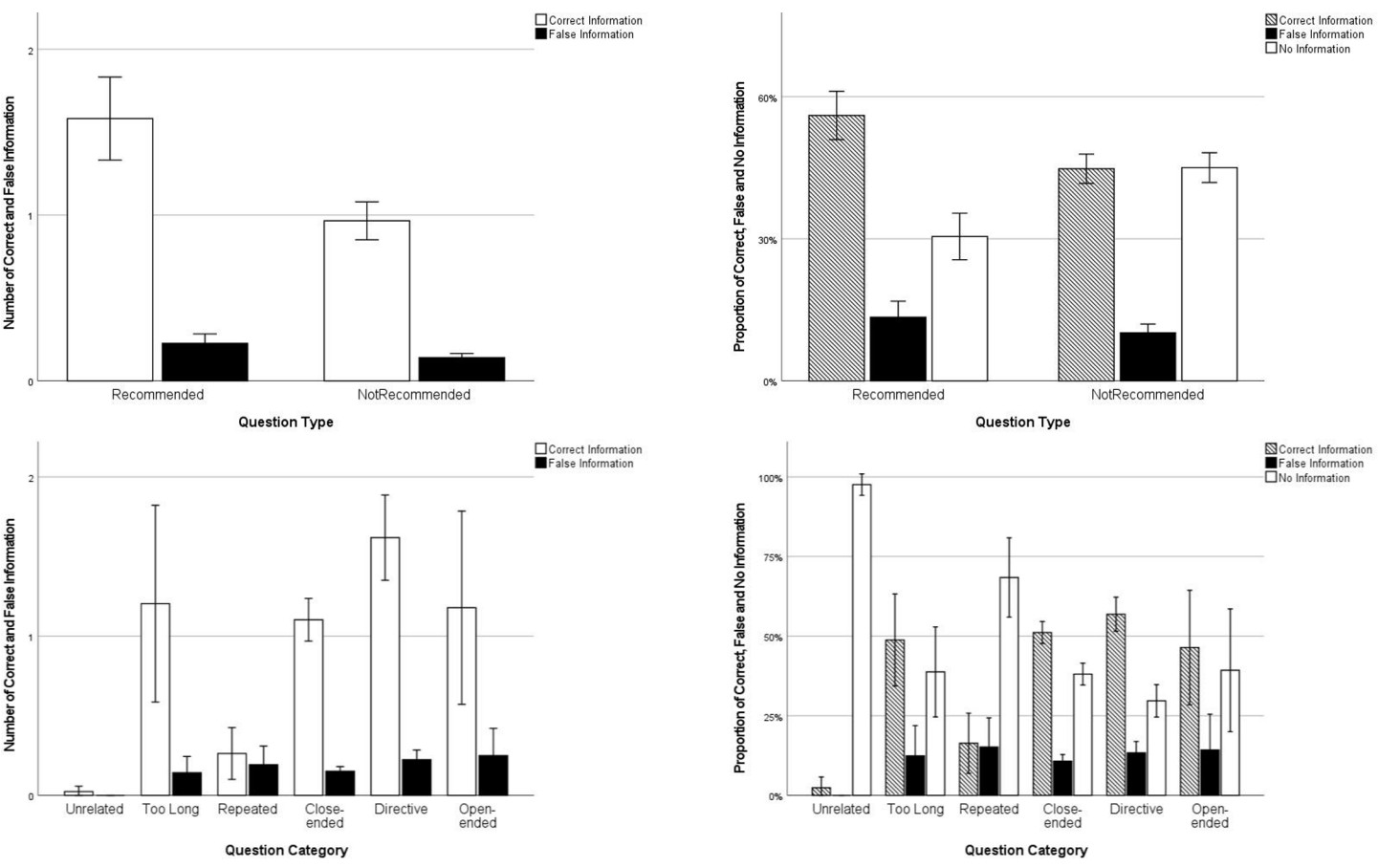

**Fig 2. Number and Proportion of Information Elicited from Different Types of Questions.**

($M$ = 0.46, $SD$ = 0.38), $F$ (1, 76) = 18.30, $p < .001$, $\eta^2$ = .19. Moreover, regarding the proportion of unique details, which indicated the unique details reported by children as a percentage of total details in the video, results showed that no difference was found between the human participants ($M$ = .10, $SD$ = .06) and LLMs ($M$ = .11, $SD$ = .05), $F$ (1, 76) = 1.05, $p$ = .310.

**Delay of question-posing by LLM and human interviews.** The delay of question-posing was defined as the time in seconds between the child finishing their answer and the interviewer starting the next question. The researchers randomly selected 30 question-answer pairs from each of the two groups and intercepted time nodes from recorded videos and calculated intervals to test whether there was any significant difference in the delay. The delay was longer for LLM ($M$ = 8.63, $SD$ = 4.91) compared to that of human interviewers ($M$ = 3.54, $SD$ = 2.50), $t$ (43.05) = 5.03, $p < .001$, $d$ = 1.30, $95\%CI$ [.74, 1.85].

**Perception of question authenticity in the LLM interview.** In the LLM group ($n$ = 40), after the interview, the researcher asked the children, "Do you think the questions just now were asked by me or computer-generated?" and recorded the children's responses. Four children answered in a voice that was too soft, so the researcher asked this question again, which might lead to social pressure. Therefore, the answers from these four children were excluded from the analyses. Among the 36 children included in the analyses, 22 children (61%) believed that the questions were generated by a real person; 14 children (39%) believed that the questions were generated by a computer. A binomial test was conducted

to test whether there was a significant difference in the authenticity, $p = .243$. Furthermore, subsequent Mann-Whitney $U$-tests showed that the difference in the number of correct information elicited from the two groups was not significant, Mann-Whitney $U = 125.50$, $p = .354$, nor in false information elicited, Mann-Whitney $U = 143.50$, $p = .728$, which means that the retrospective perception of the LLM being human or not had no effect on information elicitation.

## Discussion

The present study compared the performance of LLMs and non-expert human interviewers in posing recommended questions (i.e., directive questions and open-ended questions) to child witnesses. By asking children to watch an interactive video containing mock-events with elements that could be abusive in other contexts, followed by either human or LLM interviews, we assessed the difference in the quality of the questions posed and the information obtained. The results showed for the first time the potential of a large language model (LLM, specifically ChatGPT; OpenAI) as a tool to aid interviewers in conducting investigative interviews with children.

In this research, we found that LLM posed the same proportion of recommended questions as humans. However, LLM posed fewer recommended questions in absolute terms given that the human interviewers posed many more questions overall. Descriptive analyses revealed that some of the questions asked by LLM were too long but that in other respects, it outperformed the naive human interviewers. We assume that this difference is due to LLM being better able to adhere to the guidelines given and also due to it not asking many follow-up questions, which was something the human interviewers did. This result provides support for future training of LLMs. The results concerning information elicitation align well with this pattern: There were no differences in the overall number of unique correct pieces of information between the two groups but the LLMs elicited more such details per question and, importantly, also elicited less false information in absolute terms but not per question. These results highlight the potential of applying LLM in child interviews through aiding interviewers to formulate relevant questions adhering to best practice recommendations. Not surprisingly, and confirming a plethora of previous research [29,88], recommended question types were associated with more unique correct information elicited from the children. It is important to recognize that this association was also observed in interviews conducted by LLM interviewers. Through studies such as these, we can learn more about typical mistakes of LLMs and provide data to correct these mistakes in subsequent studies and ultimately, in real-life contexts where AI can hopefully be used to assist interviewers in their complex tasks.

LLM-generated questions performed well in terms of authenticity. In the LLM interview group, most children believed that the questions were posed by a person, and there was no significant difference in the amount of correct information elicited regardless of whether the children thought that the questions had been asked by a human or an LLM. However, it seemed that some of the children did not start paying attention to the issue of authenticity until they were asked about it. At the same time, children may not truly understand the real meaning of "computer-generated." The reason children judge that "these questions were computer-generated" could be that the concept of "computer-generated" attracts their interest. Horst et al. [89] has found that children have an endogenous bias to novelty, i.e., tend to choose things they are novel and interested in, even if they do not understand them.

In the present study, the responses of the child were transcribed and inserted into the LLM by the interviewer who then also read the next question provided by the LLM. Not surprisingly, this resulted in a difference in the delay between the child's response and the next question they were asked. This delay can be reduced in the future by automating the

speech-to-LLM and LLM-to-speech processes. However, our findings revealed that despite longer question intervals in LLM-mediated interviews compared to human interviews, LLM interviews elicited more information from children. This phenomenon can be explained through several theoretical frameworks. According to the Psychological Refractory Period (PRP) theory [90], which posits that individuals experience a brief period of reduced ability to respond to a second stimulus immediately following the processing of an initial stimulus, this delay could have allowed children to complete their cognitive processing of the previous question and prepare for the next one[91–93]. The Cognitive Load Theory [94,95] further suggests that these natural pauses might have reduced the overall cognitive burden on children's working memory, enabling more thorough information processing and retrieval[48,96,97]. The slower pace of LLM interviews may have provided an optimal cognitive processing window for the participating children, supporting their retrieval processes. This is particularly beneficial for children who, compared to adults, typically require more time for information processing and response formulation. These findings suggest that the natural delay in LLM-mediated interviews might inadvertently create more developmentally appropriate interviewing conditions for children, ultimately leading to more effective information elicitation.

While the results are encouraging, it is also evident that as used in the present study, the LLM did not yet offer the level of high-quality assistance that experienced human interviewers would be likely to find useful or that could alone provide enough assistance to a novice interviewer. Future research efforts should focus on refining and fine-tuning LLMs for these specific applications. To advance our understanding, future studies should assess LLMs' performance in various interviewing tasks and their ability to comprehend other cognitive skills relevant to investigative interviewing in addition to formulation of questions. These skills may include testing hypotheses and scrutinizing the underlying assumptions behind each allegation to prevent confirmation bias and other biases that could lead to unfair or prejudiced investigations [3,55,98,99]. It is also important to acknowledge that a high-quality investigative interview involves more than just asking evidence-based questions. Aspects such as rapport building and other elements of social interaction are equally crucial [100]. Facilitative prompts and other non-suggestive, supportive elements are known to be a good addition to open-ended questions and these might remain as the interviewer's responsibility [101]. For LLMs to be able to assist also in these aspects of the interviewing, they need to be trained to be child-friendly, considerate, and trauma-informed. It remains to be seen whether real time assistance also in this regard hampers or enhances human interviewer performance.

Moreover, it is crucial to ensure that while we aim to create tools to mitigate human factors present in interviews, we do not inadvertently transfer human-originated biases to the models we develop. Ensuring the LLMs conduct unbiased, culturally sensitive, and objective interviews will likely require iterative development, drawing on the collective expertise of legal psychologists, developmental psychologists, law enforcement, legal professionals and AI specialists. To promote transparency and collaborative progress, we advocate for open-source LLMs for forensic assistance, allowing the broader community to contribute to their development and improvement. These models must undergo testing in diverse settings, across different cultures, and with various types of children to ensure their functionality and validity in real-world investigative interviews.

Based on these preliminary results, we consider it worthwhile exploring further the potential of LLMs and artificial intelligence to enhance the consistency and quality of investigative interviews conducted by human interviewers. In the present study, we tested an LLM's ability to assist interviews with highly cooperative children who had viewed a short film depicting a neutral situation. We now need longer interviews involving children with varying levels of motivation, cognitive abilities, and ages. New developments in the LLM field involve

multimodality features of LLMs that could make it possible for a model to listen to an ongoing investigative interview and base suggestions on what it has heard. Obviously, such technology needs to be built with attention to the confidentiality and sensitivity of the data it would be processing. The way the technological landscape looks at the moment when we write this article, viable solutions for addressing these needs include online services for businesses that keep your data safe and follow privacy laws, local online services, and in the future, when more powerful AI-enabled server hardware becomes more easily available, running the program on local servers. Each of these solutions comes with unique challenges, benefits, and costs. Future research is needed to evaluate the best way forward.

The integration of Artificial Intelligence (AI) with legal investigations has undergone substantial evolution and expansion in recent years. According to Faqir [102], AI has been comprehensively integrated into criminal investigations, encompassing various aspects from arrest procedures and release decisions to sentencing processes and recidivism prediction. This widespread adoption shows AI's transformative potential in enhancing multiple dimensions of legal investigations. For example, in the domain of evidence analysis, AI technologies have showed remarkable capabilities. Machine learning techniques have significantly enhanced the analysis and organization of case data, enabling investigators to process vast amounts of information more efficiently and accurately. Moreover, in terms of linguistic analysis and credibility assessment, AI offers innovative approaches to traditional challenges. Large Language Models can automate and refine the application of established techniques such as Criteria-Based Content Analysis (CBCA) and Reality Monitoring [103]. These systems can analyze verbal statements to identify linguistic markers indicating truthfulness or deception, potentially detecting subtle inconsistencies or patterns that human analysts might overlook. Further, AI systems show promising potential for enhancing interview techniques in the realm of investigative strategy [104,105]. They can design and manage tasks or questions intended to increase cognitive load on subjects, thereby making deception more detectable. Through historical data analysis and pattern recognition, AI systems can also help law enforcement agencies more effectively predict and prevent criminal activities. Song and Li [106] suggest that the combination of large data technologies and AI can significantly enhance cyber crime prevention and investigation capabilities.

However, as Richmond [107] emphasizes, successful implementation of AI in legal investigations requires addressing three critical factors: technical reliability, ethical compliance, and legal transparency. While challenges exist, the potential benefits of AI applications in this field warrant continued development and refinement of these technologies, with appropriate attention to ethical and legal considerations.

## Limitations

The performance of out of the box LLMs depends on the prompt(s) they are given. Planning and choosing the prompt is crucial for an LLM tasked with conducting an interview in a research context because it directly impacts the quality and relevance the data gathered. A well-defined and carefully considered prompt ensures that the interview is structured, focused, and aligned with the research objectives. It helps the LLM ask pertinent questions, elicit valuable information, and maintain a coherent and logical flow during the interview. Properly addressing the prompt also reduces the risk of introducing bias or personal opinions into the interview, ensuring that the collected data remains objective and unbiased. It may be that other prompts would have resulted in another pattern of results.

It is essential to recognize the limitations of LLMs and their boundaries. Human interviewers using LLMs may still require prior knowledge and experience to ensure interview quality. Further research should investigate the dynamics of human-AI interaction during

investigative interviews performed by trained interviewers. Questions regarding how suggested information should be incorporated seamlessly into interviews without disrupting the interviewer's attention and meta-cognitive processes need exploration. Additionally, different interfaces and technical solutions for administering LLM-provided information should be thoroughly studied. Personal preferences for such assistance may vary and interviews are conducted both in the office and out in the field, making it essential to offer LLM-based aids in various technical forms. An open-source platform for development would facilitate progress in this domain.

The present study involved child participants who were interviewed immediately after viewing a brief video of a mock-event. The children displayed a high degree of cooperation and motivation in responding to the questions posed by the interviewer. It is unclear to what extent the results would generalize into actual investigative interview scenarios. Also, the video material used in this experiment was relatively simple, which may have resulted in a ceiling effect in the number of correct details obtained in both LLM and human interviews, and thus contributed to the non-significant difference. Also, while the LLM showed the ability to generate open-ended questions during these brief interviews, it became apparent that it lacked the persistence and thoroughness exhibited by human interviewers. The LLM ceased formulating questions relatively quickly once the child provided some information about the video. This indicates that, given the prompts used in the present study, the LLM was unable to determine the appropriate level of detail required or how much a child may be capable of elaborating on a specific event. Moreover, in the present study, although the prompts given to both human interviewers and LLMs employed advanced technology such as instruction tuning [67] and few-shot learning [68,70,71], it contained only abstract requirements rather than specific guidelines (such as NICHD protocols). As such, our findings solely reflect the baseline capabilities of LLMs. Although LLM performance is context-dependent, it remains an open question whether LLMs could effectively adhere to professional guidelines when provided with comprehensive professional instructions.

Research conducted by Powell et al. [28] found that experience in investigative interviews had a negative impact on the usage of open-ended questions. Moreover, Lamb [27] argued that even trained interviewers do not use as many appropriate questions as recommended. Given that the role of experience in quality of question formulation is not clear and given that our study was the first one to assess LLMs' performance in this domain, we decided to include naive human interviewers. One of the outcomes of the study could have been that LLMs perform clearly worse than naive human interviewers which would make it unnecessary to compare their performance with experienced interviewers. However, we believe that the results evidence LLMs having some strengths in this task suggesting that future comparisons should include experienced human interviewers trained to use structured interview protocols that have been proven to improve human performance.

Finally, the LLM exhibited a tendency to pose questions and make statements that were sometimes overly lengthy for child victims or witnesses of abuse. Long or multiple questions are often too difficult for children to understand properly [108]. Even human experts and professionals working with children find it challenging to employ age-appropriate language, so it is not surprising that the LLM requires further development to enhance its performance in this regard [88].

In the present study, the participating children witnessed a neutral/funny mock-event; they did not personally experience or witness traumatic events. According to Waterhouse [109], establishing rapport with children who experienced non-traumatic events may not yield better outcomes than not doing so. Furthermore, research by Daviesl et al. [110], Teoh and Lamb [111], and Roberts et al. [112] suggests that spending excessive time on establishing rapport

can diminish interview efficiency, potentially due to fatigue experienced by children during the interview process. Therefore, in the prompts of our study, we did not require human interviewers or LLMs to engage in rapport-building with children. Furthermore, we conducted a series of analyses after removing the rapport questions such as "What's your name?", "How old are you?" or "Do you like to play Lego chess?". The result showed that there's no difference between the number of questions, the number of types of questions, the number of absolute or proportion of correct and false information elicited between the data containing rapport questions and without such questions.

In future research, the guiding information given to the LLM can be further modified with clear requirements to reduce the proportion of undesirable questions and enable deeper probing into the details of children's answers. The age of children was restricted to 6-8 years. Since children's memory develops rapidly, the research conclusions cannot be generalized to all children, especially those whose memory has not fully developed.

The real human interview volunteers recruited for the study could only simulate the level of untrained human interviewers. In future studies, trained human interviewers could be recruited for comparison. More experts or psychologists could be recruited in this topic of studies to test whether experience or knowledge in this field could have significant effects in the future.

During child interviews, the instructions did not specify requirements related to relationship building. Although some human volunteers showed relationship-building behaviors, such as discussing the child's preferences before the official interview, most human volunteers and all LLM interviews did not reflect this aspect, diving directly into the interview. This does not simulate the real-life interrogation process. Moving forward, improvements and modifications should be made to the instructions, ensuring that the interviewer establishes a stable relationship before formally starting the interview process.

## Conclusion

Our findings suggest, for the first time, that LLMs have potential in serving as investigative interview support tools. The results also point to challenges in applying them in their current form, wherefore it is necessary to fine-tune LLMs in order to solve these challenges. In future research, we aim to delve deeper into the capabilities of LLMs in the context of conducting interviews with children. Should the outcomes remain promising, LLMs could be used to support human interviewers in real time by providing suggestions for context-appropriate and open questions. LLMs would analyze the child's responses during the ongoing interview and would immediately propose the next question to the interviewer. These suggested questions could be presented to the interviewer through different interfaces, such as pads, computer screens placed behind the child or via Augmented Reality glasses. This approach would have the potential to bolster the effectiveness of human interviewers. Also, the LLMs role would be to offer question suggestions with the human interviewer autonomously deciding whether to use the suggestion or reject it and ask another question instead. This means that the responsibility for the questions asked would squarely be that of the human interviewer. A future possibility would be to replace human interviewers with completely autonomous avatar or robot interviewers. However, this requires intensive investigations into potential differences in rapport building between artificial agents and humans. Such an approach may also not be legally possible at this time.

## Supporting information

**S1 Table. Content of the Mock-Event Videos.**
(DOCX)

**S2 Table. Coding of Static and Dynamic Elements in Child Answers**
(DOCX)

**S3 Table. Interview Transcripts for the Most Typical LLMs and Human Interviewer Interviews.**
(DOCX)

## Author contributions

**Conceptualization:** Haohai Pang, Pekka Santtila.

**Data curation:** Yongjie Sun, Pekka Santtila.

**Formal analysis:** Yongjie Sun, Pekka Santtila.

**Funding acquisition:** Liisa Järvilehto, Pekka Santtila.

**Investigation:** Yongjie Sun, Haohai Pang, Ophelia Zhang.

**Methodology:** Yongjie Sun, Haohai Pang, Liisa Järvilehto, David Shapiro, Shumpei Haginoya, Pekka Santtila.

**Project administration:** Liisa Järvilehto, Pekka Santtila.

**Resources:** Haohai Pang, Ophelia Zhang, Pekka Santtila.

**Software:** Yongjie Sun, David Shapiro.

**Supervision:** Liisa Järvilehto, Julia Korkman, Pekka Santtila.

**Validation:** Yongjie Sun.

**Visualization:** Yongjie Sun, Pekka Santtila.

**Writing – original draft:** Yongjie Sun, Liisa Järvilehto, Pekka Santtila.

**Writing – review & editing:** Yongjie Sun, Haohai Pang, Liisa Järvilehto, Ophelia Zhang, David Shapiro, Julia Korkman, Shumpei Haginoya, Pekka Santtila.

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
