## [Decision Letter · Decision Letter 0]

13 Nov 2024

PONE-D-24-16051

Comparing the performance of a Large Language Model and naive human interviewers in interviewing children about a witnessed mock-event

PLOS ONE

Dear Dr. Sun,

Thank you for submitting your manuscript to PLOS ONE. After careful consideration, we feel that it has merit but does not fully meet PLOS ONE’s publication criteria as it currently stands. Therefore, we invite you to submit a revised version of the manuscript that addresses the points raised during the review process.

We look forward to receiving your revised manuscript.

Kind regards,

Primož Kocbek, BSc

Academic Editor

PLOS ONE

Journal Requirements:

3. Thank you for stating the following financial disclosure: “This work was supported by the Sundell Foundation and the Shanghai Frontiers Science Center of Artificial Intelligence and Deep Learning at NYU Shanghai.”

4. Please note that your Data Availability Statement is currently missing the repository name and/or the DOI/accession number of each dataset OR a direct link to access each database. If your manuscript is accepted for publication, you will be asked to provide these details on a very short timeline. We therefore suggest that you provide this information now, though we will not hold up the peer review process if you are unable.

7. Please remove your figures from within your manuscript file, leaving only the individual TIFF/EPS image files, uploaded separately. These will be automatically included in the reviewers’ PDF**.**

9. We note that this data set consists of interview transcripts. Can you please confirm that all participants gave consent for interview transcript to be published? If they DID provide consent for these transcripts to be published, please also confirm that the transcripts do not contain any potentially identifying information (or let us know if the participants consented to having their personal details published and made publicly available). We consider the following details to be identifying information: - Names, nicknames, and initials - Age more specific than round numbers - GPS coordinates, physical addresses, IP addresses, email addresses - Information in small sample sizes (e.g. 40 students from X class in X year at X university) - Specific dates (e.g. visit dates, interview dates) - ID numbers Or, if the participants DID NOT provide consent for these transcripts to be published: - Provide a de-identified version of the data or excerpts of interview responses - Provide information regarding how these transcripts can be accessed by researchers who meet the criteria for access to confidential data, including: a) the grounds for restriction b) the name of the ethics committee, Institutional Review Board, or third-party organization that is imposing sharing restrictions on the data c) a non-author, institutional point of contact that is able to field data access queries, in the interest of maintaining long-term data accessibility. d) Any relevant data set names, URLs, DOIs, etc. that an independent researcher would need in order to request your minimal data set. For further information on sharing data that contains sensitive participant information, please see: https://journals.plos.org/plosone/s/data-availability#loc-human-research-participant-data-and-other-sensitive-data If there are ethical, legal, or third-party restrictions upon your dataset, you must provide all of the following details (https://journals.plos.org/plosone/s/data-availability#loc-acceptable-data-access-restrictions): 1. A complete description of the dataset 2. The nature of the restrictions upon the data (ethical, legal, or owned by a third party) and the reasoning behind them 3. The full name of the body imposing the restrictions upon your dataset (ethics committee, institution, data access committee, etc) 4. If the data are owned by a third party, confirmation of whether the authors received any special privileges in accessing the data that other researchers would not have 5. Direct, non-author contact information (preferably email) for the body imposing the restrictions upon the data, to which data access requests can be sent

Reviewers' comments:

Reviewer's Responses to Questions

**Comments to the Author**

1. Is the manuscript technically sound, and do the data support the conclusions?

Reviewer #1: Partly

Reviewer #2: Partly

2. Has the statistical analysis been performed appropriately and rigorously? 

Reviewer #1: No

Reviewer #2: I Don't Know

3. Have the authors made all data underlying the findings in their manuscript fully available?

Reviewer #1: Yes

Reviewer #2: Yes

4. Is the manuscript presented in an intelligible fashion and written in standard English?

Reviewer #1: Yes

Reviewer #2: Yes

5. Review Comments to the Author

Reviewer #1: I am in favor of the publication of this manuscript in PLOS ONE. However, I request modifications to enhance the impact and relevance of this study. This will encourage further research in this field and lead the authors and other teams to continue this (very relevant) work. The subject is relevant, interesting, and well-suited to our contemporary times. The involvement of AI in conducting interviews is a fundamental aspect that is neglected and very rarely explored by scientists. The questions raised by this team of researchers are intriguing and lead to new questions that, I hope, will find answers soon. In other words, this article represents a source of inspiration and motivation for contemporary research in forensic psychology applied to interview methods.

The work presented here involved participants who are not professional interviewers. This might seem like a weakness of the study, as one might expect the AI's performance to be compared to that of law enforcement officers. However, it is often overlooked that most crimes against children are not discovered by professional investigators but, for example, by kindergarten or primary school teachers, social workers, neighbors, etc. The comparison group used by the authors in their study is therefore relevant because it addresses a social issue: is it better to use AI to listen to a child who is suspected of being a victim of abuse or violence, or should we allow the person who becomes aware of the abuse to interview the child, even if they have never been trained to perform this difficult task?

Overall, the article is well-structured and organized. The abstract is clear and accurately reflects the content of the article. However, it might be helpful to include in parentheses what 'Recommended' and 'Not Recommended' questions are, as readers who are not specialists in this field (law enforcement officers, social workers, doctors, journalists, policymakers, etc.) may not immediately understand what these terms describe. A sentence could be: "Irrespective of the condition, recommended questions (XXX, XXX) elicited more correct information than not recommended ones (YYY, YYY, YYY, and YYY)."

The keywords are relevant. Perhaps a keyword related to 'questions,' which are the main measure of the article, should be added.

Overall, the literature review, the research question, the description of the materials and procedure are well-written, relevant, and meet expectations. However, I am more cautious regarding the data analysis (results) and its description. The discussion is interesting and covers all the aspects addressed. The future directions and limitations are relevant and prompt the reader to reflect on what should be done in terms of experimental research and practical application in the field.

Here are some comments and remarks.

____ INTRODUCTION ____

P.4, L.78___ The description of international recommendations on the use of open-ended questions is insufficient. An uninformed reader who is not familiar with the topic will not understand what should be done and why. Examples of different types of questions are necessary, and these examples should be supported by data from the scientific literature. This literature can include studies on interview methods with witnesses and victims in general (see, for example, research on cognitive interviews, which contains many recommendations suitable for this manuscript), but especially those related to the NICHD. These elements are fundamental and should be addressed in a dedicated paragraph. A more detailed section on the NICHD and the necessity of its various phases should also be written, as it will help conceptualize future research and clarify the discussion.

P.4, L.95 & L.100 ___ 'individual differences'  this expression is unclear, and it is not clear whether it refers to the interviewees, the interviewers, or someone else. Not indicating this forces the reader to refer to the original article (which is not a major issue in itself, but it disrupts the flow of reading and, therefore, the strength of the authors' argument).

P.5, L.116 and following ___ This section is interesting, and the arguments presented are relevant and appropriate. Reading it made me think of the Self-Administered Interview (SAI), which does not require the intervention of a professional. It could be interesting to add a few sentences highlighting the strengths and weaknesses of this method compared to the use of AI and NICHD-type interviews (even though I am fully aware that the SAI has not been tested with children). The concept of Self-Report used in the SAI is also addressed later in this manuscript, P.6, L.151.

____ AIMS AND HYPOTHESES ____

I generally agree with the three hypotheses put forward by the authors, but they should be justified by referencing key findings from the literature that support their claims. It is essential that the reader does not have to search for information and reconstruct the authors' argument and reasoning for their hypotheses.

Without any theoretical justifications, I could formulate opposing hypotheses based on a different line of reasoning: if I understand the functioning of LLMs correctly (I am not an expert in this field), these algorithms 'learn' a task or function by processing large amounts of data. In this case, CHAT GPT was trained on interviews of standard quality. In other words, the training data likely did not include best practices. For example, the training interviews probably contained many more 'not recommended' questions than 'recommended' ones. Therefore, how could CHAT GPT produce 'recommended' questions if it rarely encountered them during training? Based on this alternative (and perhaps completely incorrect and irrelevant) reasoning, we could expect, at best, 'equality' between humans and the machine (which is, of course, an invalid hypothesis), or at worst, a lower quality of questions posed by the machine compared to those posed by humans.

____ METHOD ____

No comment

____ PARTICIPANTS ____

The authors conducted and presented an a priori power calculation, which is relevant and appreciated. However, the effect size proposed (and assumed) by the authors (i.e., .500) seems to come out of nowhere. Without any justification, this gives the informed reader the impression that this calculation was included in the manuscript to justify a sample size constrained by reality (i.e., the authors only had access to the mentioned sample size and chose the effect size that would justify their imposed sample)

___ MATERIALS ____

No Comment

____ PROCEDURE ____

My comments on the procedure mainly concern the instructions (prompts) given to the LLM.

For example, these instructions do not mention the main NICHD recommendations, even though the coding variables used and presented on the following page are drawn from a meta-review on the NICHD. For instance, it is essential that the interviewee be allowed to give a 'free recall' following a very general prompt inviting the child to recall everything they remember. And the child should not be interrupted. These few guidelines are fundamental and distinguish a good interview from a bad one. I mention this so that the authors justify the absence of such instructions. In fact, readers who are well-versed in this specific topic will immediately notice the absence of mentions on these points. However, I know and clearly understood from reading the procedure that the children were never interrupted while speaking, whether by the human or the machine.

Table 2 mentions in its last paragraph that both the machine and the human should obtain a description of the adults present in the video. However, no data concerning the physical appearance of the individuals is presented in the manuscript. At most, we find mentions of certain 'dynamic' elements on P.14, L.309 and following, as well as in Table 5.

____ CODING ____

I believe that the choice of coding variables is one of the contentious points of this manuscript. For instance, I understand that correct, incorrect, and confabulations are sometimes redundant informations (repeated multiple times by the same child) are counted each time as a "new" information. As a result, following this coding approach, a child who repeats 20 times that there was a man in the video will receive a score of 20 correct pieces of information, which is not consistent with the established literature. This coding method skews the results and, consequently, the interpretation of performance, which will appear far superior to what it actually is. A reported piece of information, whether correct or incorrect, should only be counted once. Saying 20 times that the adult was a man provides the same amount of information as saying 'it was a man' just once.

Moreover, the Accuracy Rate variable, which is frequently mentioned in the literature on judicial interviews, is missing. The accuracy rate should be calculated and statistically analyzed. It is calculated by dividing the number of UNIQUE correct pieces of information by the total number of UNIQUE pieces of information reported. This information is crucial and will help better interpret and understand the results of the analyses, which are sometimes contradictory or uninterpretable. I know I am asking a lot of the authors and that this will slow down the publication process (as I hope the editorial committee will ultimately decide to publish this manuscript), but the investment placed in obtaining this verbal data cannot be diminished due to an unstudied variable. Table 6 should therefore be modified to now include this new variable. Additionally, the authors can keep the 'cumulative' volumes of correct information, errors, and fabrications, but they must also report the volumes of UNIQUE information and, unfortunately, redo the analyses accordingly.

Tables 4 and 5 could be moved to the appendices, as they provide little information relevant to understanding the authors' argument.

____ CODING OF QUESTIONS AND ANSWERS ____

Information about the coders is needed. How many were there? etc.

____ STATISTICAL ANALYSES ____

It is not correct to perform multiple t-tests when dependent variables are most likely correlated with each other. Authors should use MANOVAs to analyze their data. All the most relevant indices should be presented in the text. Another interesting piece of information would be to report the correlations between the different variables in a correlation matrix. This information will make it easier to understand and interpret the sometimes contradictory results in this version of the manuscript.

I have noticed that the statistical values reported in the first two analyses (P. 21, L.379-384) are exactly the same (t, p, d, 95% CI). This may be a coincidence, but it's worth checking.

The order of the dependent variables analyzed could be modified. Generally, we start with correct information, errors and affabulations, total and accuracy rate. Please note that this information is sometimes highly correlated, like correct information and accuracy rate. Care should be taken when presenting and discussing the results. The total volume of information and the accuracy rate are conceptually independent and should be the focus of the authors' discussion.

Certain terms, such as “marginally” (P. 22, L. 409), "significantly" (P.23, L. 436)and “partially” (P. 21, L. 384), should not be used in the presentation of results, as they could be misinterpreted by readers who are not specialists in statistics and its subtleties.

Tables 1 and 7 could be placed in an appendix, as they do not contain information crucial to understanding the authors' arguments.

Decimals in numerical values should be two, with the exception of the p value.

Graphs showing total volumes of information and accuracy rates as a function of question type would be relevant and facilitate understanding of the phenomenon under study.

Table 10 could be structured around the two categories “recommended/Not recommended”. These should appear clearly in Table 10. It would also be relevant for a table like Table 10 to present all the means and standard deviations of the different variables according to both question type and origin (Human vs. LLM).

____ DISCUSSION ____

The discussion in its current form is interesting and takes up the results. Many of the comments are pertinent, particularly those concerning the study's limitations. The authors are genuinely self-critical, which will serve to improve their initial paradigm and thus, ultimately, the relevance of their work. However, I think it will have to be modified in the light of the results of the new analyses to be carried out.

___ MINOR COMMENTS ___

P.4, L.93 ___ mega-analysis ? - meta-analysis ?

Reviewer #2: 1. The objectives and hypotheses could be stated more clearly at the beginning of the manuscript.

2. While a power analysis was conducted, it would be beneficial to provide more detail on how the sample size was determined and whether it was sufficient to detect meaningful differences.

3. More information on the randomization process for assigning children to either the LLM or human interviewer condition would strengthen the methodology section. Clarifying how randomization was implemented can help assess the validity of the results.

4. The interaction between the LLM and the interviewer could be elaborated upon. For instance, detailing how the prompts were structured and the rationale behind the specific instructions given to both the LLM, and human interviewers would provide insights into the experimental design.

5. he delay in question posing by the LLM compared to human interviewers is an interesting finding. However, the implications of this delay on the flow of conversation and the children's responses should be discussed in more depth.

6. While t-tests and Mann-Whitney U tests were employed, a more detailed explanation of the statistical methods used, including assumptions and justifications for their use, would enhance the rigor of the analysis. Additionally, reporting effect sizes for all significant findings would provide a clearer understanding of the practical significance of the results.

7. The distinction between incorrect and confabulated information is crucial. A more thorough explanation of how these categories were defined and coded would improve the clarity of the findings.

8. The discussion could benefit from a more extensive exploration of how the findings relate to existing literature on child interviewing and the use of AI in investigative contexts. This would help situate the study within the broader field and highlight its contributions.

6. PLOS authors have the option to publish the peer review history of their article (what does this mean? ). If published, this will include your full peer review and any attached files.

**Do you want your identity to be public for this peer review?** For information about this choice, including consent withdrawal, please see our Privacy Policy .

Reviewer #1: **Yes: ** Samuel DEMARCHI

Reviewer #2: No

---

## [Author Response · Author response to Decision Letter 0]

20 Nov 2024

Dear Editor and Reviewers,

We sincerely thank you for your thorough and constructive feedback on our manuscript. Your comments have helped us significantly improve both the clarity and rigor of our work. We have carefully addressed each point raised and made substantial revisions accordingly.

Following your recommendations, we have enhanced our statistical analysis by implementing MANOVA instead of multiple t-tests, given the identified correlations among dependent variables. We have added correlation matrices for primary variables and reported comprehensive effect sizes for all significant findings. We have also introduced "Proportion of Unique Information" as a new variable to better capture the nature of children's responses.

We have strengthened our methodological framework by providing detailed explanations of participant randomization procedures and power analysis rationale. The coding procedures have been clarified with comprehensive examples distinguishing between correct, incorrect, and confabulated information. To better explain our findings, we have integrated Psychological Refractory Period (PRP) theory and Cognitive Load Theory, particularly in relation to interview delay effects. We have also expanded our discussion of LLM capabilities, focusing on instruction tuning and few-shot learning, while providing detailed justification for including naive participants as a comparison group.

The manuscript's structure has been improved through reorganized results presentation, enhanced figures, and standardized numerical formatting. Supplementary tables have been moved to supporting information to maintain focus on key findings. All modifications are highlighted in blue in the track-changes version, and both track-changes and clean versions have been uploaded.

We believe these revisions have substantially strengthened our manuscript and hope it now meets your requirements for publication.

Sincerely,

The Authors

---

## [Decision Letter · Decision Letter 1]

10 Dec 2024

Comparing the performance of a Large Language Model and naive human interviewers in interviewing children about a witnessed mock-event

PONE-D-24-16051R1

Dear Dr. Sun,

We’re pleased to inform you that your manuscript has been judged scientifically suitable for publication and will be formally accepted for publication once it meets all outstanding technical requirements.

Kind regards,

Primož Kocbek, BSc

Academic Editor

PLOS ONE

Additional Editor Comments (optional):

Note that there is a final comment from reviewer 1 for your consideration, to exchange the term "demonstrate" with "show" in the manuscript (check reviewer 1 revision 1 comments for more details).

Reviewers' comments:

Reviewer's Responses to Questions

**Comments to the Author**

1. If the authors have adequately addressed your comments raised in a previous round of review and you feel that this manuscript is now acceptable for publication, you may indicate that here to bypass the “Comments to the Author” section, enter your conflict of interest statement in the “Confidential to Editor” section, and submit your "Accept" recommendation.

Reviewer #1: All comments have been addressed

Reviewer #2: All comments have been addressed

2. Is the manuscript technically sound, and do the data support the conclusions?

Reviewer #1: Yes

Reviewer #2: Yes

3. Has the statistical analysis been performed appropriately and rigorously? 

Reviewer #1: Yes

Reviewer #2: Yes

4. Have the authors made all data underlying the findings in their manuscript fully available?

Reviewer #1: Yes

Reviewer #2: (No Response)

5. Is the manuscript presented in an intelligible fashion and written in standard English?

Reviewer #1: Yes

Reviewer #2: Yes

6. Review Comments to the Author

Reviewer #1: First of all, I would like to thank the authors for the quality of their responses and the explanations provided in their letter to the reviewers. While I do not necessarily agree with some of the modifications or responses, I am confident that our differences will be of interest to the scientific community.

However, I have one final comment: the term "demonstrate" is used in various forms multiple times in the revised manuscript. This term is not appropriate because experimental science does not, under any circumstances, allow for the proposal of a demonstration. Therefore, all occurrences should be reviewed and replaced, for example, with the verb "show," which is a more versatile and entirely suitable formulation for an experimental study.

Reviewer #2: The manuscript adequately addresses research ethics, as it includes IRB approval (2022-035-NYUSH-New Bund) and outlines informed consent processes for both parents/guardians and child participants. Additionally, the ethical considerations for interviewing children and safeguarding their rights during the study are well-documented.

There do not appear to be concerns regarding dual publication, as the manuscript is presented as original research, with no indication that it has been published elsewhere or submitted concurrently to another journal. However, I recommend verifying this to ensure compliance with publication standards.

Regarding publication ethics, the authors have transparently disclosed funding sources, declared no competing interests, and made data publicly available. These practices align with ethical publication guidelines. A final check for the originality of the analyses and any potential overlap with prior publications by the authors would be prudent to confirm adherence to ethical standards.

7. PLOS authors have the option to publish the peer review history of their article (what does this mean? ). If published, this will include your full peer review and any attached files.

**Do you want your identity to be public for this peer review?** For information about this choice, including consent withdrawal, please see our Privacy Policy .

Reviewer #1: **Yes: ** Samuel DEMARCHI (Ph.D)

Reviewer #2: No

---

## [Editor Report · Acceptance letter]

PONE-D-24-16051R1

PLOS ONE

Dear Dr. Sun,

I'm pleased to inform you that your manuscript has been deemed suitable for publication in PLOS ONE. Congratulations! Your manuscript is now being handed over to our production team.

Kind regards,

on behalf of

Mr. Primož Kocbek

Academic Editor

PLOS ONE